# Evaluation of the Accuracy of Pattern Recognition by a Neural Network with Various Filters in the Receptor Layer of the Retinal Simulation Module

## Abstract

The purpose of this work is to evaluate the effect of the location of receptors in the first layer of the retinal simulation module on the ability of a neural network to recognize images. The retinal simulation module serves as a means for preprocessing images. The retinal simulation module is described and compared with existing popular preprocessing methods. The module processes the image using three layers. The object of this study is the first layer of the module, which simulates the receptor layer of the real retina of the human eye. The experiments were conducted on a fully connected neural network. The retinal simulation module preprocessed a sample of fruit images photographed from different angles, which was then fed to the input of the neural network. In the process, ninety-four experiments were performed with different module settings. In each of the experiments, the settings of the fully connected neural network remained unchanged. The results of image recognition by a neural network are presented. Recommendations are given for configuring the receptor layer of the retinal simulation module to improve the accuracy of pattern recognition.

## 1 Introduction

In many branches of human activity, such as medicine, construction, transport, etc. (Tupileikina, 2023; Sakulin and Alfimtsev, 2024; Korotysh et al., 2025; Lokteva et al., 2022), an urgent task is to obtain information important for processes and automate activities in general (Vidmanov and Alfimtsev, 2024a;b; Bolshakov et al., 2024), which is possible, among other things, by introducing visual recognition of objects into their control and monitoring systems and further displaying information using the means augmented reality, which will eventually affect the timely response of employees and improve their work safety.

Existing image recognition methods and algorithms are often based on natural information perception mechanisms, including the human perceptual system, which includes the visual system, consisting of such basic elements as the retina, optic nerve, and visual cortex. Research and analysis of such a human system that visualizes the environment is still relevant and may lead to the emergence of new methods and algorithms for detecting and recognizing objects (Adamova et al., 2021; Loktev et al., 2023), determining object parameters (Loktev et al., 2022), building virtual/augmented reality, etc. To do this, this paper examines the retinal simulation module and analyzes the effect of the location of receptors in the module on the ability of a neural network to recognize images.

## 2 Materials and methods

Since this paper examines the effect of the location of receptors on the retinal layer, it is necessary to consider the literature in which the receptor layer was modeled or images of the real retina of the eye were used.

The article (Lebedev and Marshak, 2007) evaluated the effect of amacrine cells using a model based on a macaque retina image. The model predicts that the enhancement of red-green antagonism is

one of the important functions of amacrine cells in the pathway providing input to dwarf ganglion cells in the retina of primates.

In (Utrobin, 2010a;b; 2013), the architectural aspects of the organization of the retina of the eyeball are considered, and variants of the structural and functional organization of the retina are presented. These articles present the possible location of two groups of retinal receptors (cones and rods). on a rectangular plane for computer vision systems, but the cone is considered as a receptor of only one type, that is, the relative arrangement of cone types reacting to different wavelengths is not described.

In the article (Momiji et al., 2005), a cellular model of the primate retina was developed. In this work, cones of only two types are considered and distributed randomly in the retina, the rods were not modeled at all.

The photoreceptors of the real retina are densely located to each other and form a hexagonal (hexagonal) lattice (Chuprov et al., 2021; Ahnelt, 1998). The mosaic of cones and rods is fixed, but the receptors are unevenly distributed across the retina (Ahnelt, 1998; Rozhkova et al., 2016; Jones and Higgins, 1947).

The articles (Roorda and Williams, 1999; Alekseenko, 2019) revealed that the ratio of M- and L-type cones differs significantly in different people with normal vision (for example, the values of 75.8% of L-type with 20.0% of M-type versus 50.6% of L-type with 44.2% of M-type in two men).

In the considered retinal models, the authors either modeled the receptor layer according to the coordinates of cones and rods from real retinal images (for example, in (Lebedev and Marshak, 2007) a section of the retina of a macaque was taken), or positioned them randomly (Momiji et al., 2005), focusing on other layers of the retina, no other approaches were found. The use of the retina model as a means of image preprocessing to improve image recognition by neural networks has not been described in publications at the moment.

## 3 DESCRIPTION OF THE RETINAL SIMULATION MODULE

The retinal simulation module consists of a layer of photoreceptors, a layer of bipolar cells, and a layer of ganglion cells.

For the module, the pixel intensity of the input image is analogous to the streams of light beams entering the receptors of the real eye. Therefore, each pixel must correspond to each "cell" of the first layer of the software retina.

The software cones of the real eye are divided into three groups according to the length of radiation they can absorb. Since the retina simulation module reads pixels from an image that are encoded using the RGB model, the software cones take the intensity value from a specific color channel of the image.

The sticks are simply excited in the presence of a certain level of light, so in the module they must convert a pixel to shades of gray and transfer the value already obtained to other layers of the software retina.

During the development of the module, pixel distribution matrices on the camera matrix, such as the Bayer filter, RGBW, and X-Trans, were used as a law modeling the location of receptors in the real retina.

As a result, the filters shown in Figure 1 were prepared for the module. They have been divided into categories based on certain distinctive features of these filters.

Each of the cells in the next layers of the module has a receptive field. In the real retina of the eye, cells have two variants of receptive fields: on- and off-type. In the on-type, the center of the receptive field excites the neuron, and the periphery slows it down; in the off-type, the opposite is true: the center slows down, and the periphery excites (Izmailov et al., 1989).

In this work, we will use a variant of the receptive fields of cells that mimics the work of the fields of real retinal cells, shown in Figure 2.

The output of a software bipolar ON-type neuron was calculated using the formula (1):

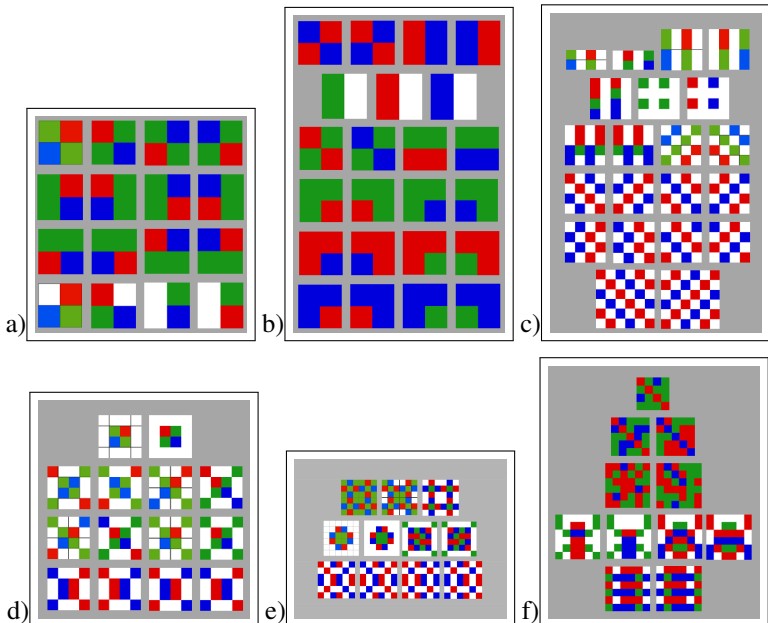

Figure 1: Filters for first layer of retinal simulation module. a) like Bayer b) two colors c) lines d) like "X" e) like XTrans f) extras.

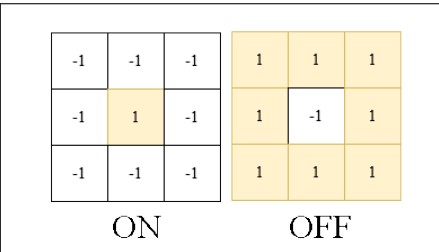

Figure 2: Organization of receptive fields of bipolar and ganglion cells of the retina, implemented in software form.

$$bip_{on} = v(x_{stim}, y_{stim}) + \sum_{k=0}^{K} (-1) * v(x\_inhib_k, y\_inhib_k) \tag{1}$$

where $bip_{on}$ is the output of an ON-type bipolar cell;

$v(x_{stim}, y_{stim})$ is the brightness value of the image pixel located in the center of the receptive region of the bipolar cell (the orange area of the ON-cell in Figure 2);

$K$ is the number of image pixels located on the periphery of the receptive field of the cell;

$v(x\_inhib_k, y\_inhib_k)$ is the brightness value of the $k$th pixel of the image located at the edge of the area of the bipolar cell (the white area of the ON-cell in Figure 2).

$$bip_{off} = (-1) * v(x_{inhib}, y_{inhib}) + \sum_{k=0}^{K} v(x\_stim_k, y\_stim_k) \tag{2}$$

where $bip_{off}$ is the output of an OFF-type bipolar cell;

$v(x_{inhib}, y_{inhib})$ is the brightness value of the image pixel located in the center of the receptive region of the bipolar cell (the white area of the OFF-cell in Figure 2);

$K$ is the number of image pixels located on the periphery of the receptive field of the cell;

$v(x\_stim_k, y\_stim_k)$ is the brightness value of the $k$th pixel of the image located at the edge of the area of the bipolar cell (the orange area of the OFF-cell in Figure 2).

The relative arrangement of cells with different receptive fields in the layer used in this work is shown in Figure 3.

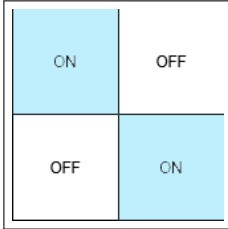

Figure 3: Arrangement of cells with different receptive fields on the layer.

For the ganglion cell layer, the same variants of the organization of the receptive fields of ON-type and OFF-type cells and the same variants of the arrangement of ON-type and OFF-type cells in the layer are used.

From formulas 1 and 2, it can be seen that the algorithm that allows calculating the output values for a layer of bipolar cells should have a time complexity of $O(n^4)$, since it is necessary to traverse the image with a window of a certain size in 1 pixel increments, therefore, the retinal simulation module uses the same approach to optimize calculations as in modern implementations of convolutional neural networks using the convolution theorem. This approach has been modified for the module, since it is necessary to get the values for ON cells using one filter, and for OFF cells using another, and then arrange them in the desired order on the final image.

## 4 COMPARISON OF THE RETINAL SIMULATION MODULE WITH OTHER IMAGE PREPROCESSING METHODS

To compare the retinal simulation module with other preprocessing methods, a sample was used, consisting of photographs of fruits photographed from different angles, measuring 100 by 100 pixels. It contains 39 classes of fruits. The training sample contains 18505 images, and the test sample contains 6215.

During the comparison, a modified X-Trans filter was installed in the receptor layer in the module, shown in Figure 4.

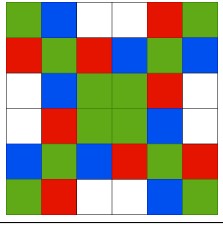

Figure 4: Modified X-Trans filter.

A new sample was made from the original sample, obtained by passing images from the original sample through the retina simulation module with the previously specified layer settings. The original sample and the one obtained using the retinal simulation module are shown in Figure 5.

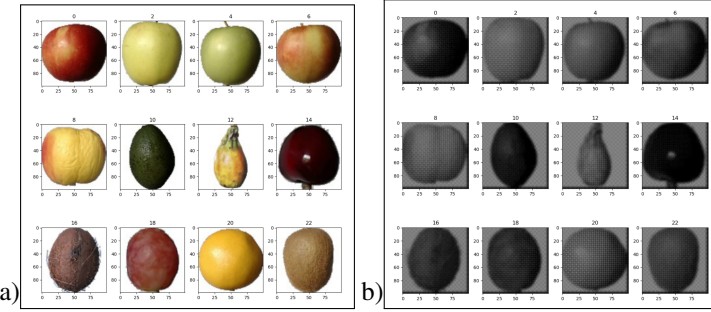

Figure 5: Samples: a) the original sample, b) an example of a sample obtained after conversion by the module of the original sample.

For pattern recognition, a fully connected neural network was created in Python (Popov, 2023) using the keras library. The resulting neural network consists of an input layer that accepts images measuring 100 by 100 pixels, and four fully connected layers. The last layer has 39 neurons, which corresponds to the number of object classes in the sample. The complete network architecture is shown in Figure 6.

```
Model: "sequential"

Layer (type)                Output Shape              Param #
=================================================================
flatten (Flatten)           (None, 10000)             0

dense (Dense)               (None, 512)               5120512

dense_1 (Dense)             (None, 256)               131328

dense_2 (Dense)             (None, 128)               32896

dense_3 (Dense)             (None, 39)                5031
=================================================================
Total params: 5,289,767
Trainable params: 5,289,767
Non-trainable params: 0
_________________________________________________________________

None
```

Figure 6: Architecture of the fully connected neural network used.

Stochastic gradient descent with Nesterov moments with a learning rate of 0.01 was chosen as the optimization method for training. The categorical cross-entropy is established as a loss function. The package size was 32 images, the number of epochs was 10.

From the training sample, 20% was allocated to the validation sample, so the neural network was eventually trained on 14,804 images. The size of the validation sample was 3,701 images.

Since the neural network starts training with random weights, 10 neural network trainings were done from scratch for each of the preprocessing methods. The results of comparing the effect of the retinal simulation module on the accuracy of neural network image recognition with other image preprocessing methods are shown in Table 1.

## 5 CONDUCTING AN EXPERIMENT

For each filter option shown in Figure 1, a sample was made from the original fruit sample used in the previous section. The architecture of the neural network and its settings did not change during the experiment. The neural network image recognition results for each of the filters are shown in Table 2.

Table 1: Comparison of the accuracy of image recognition by a fully connected neural network with different methods of preprocessing on a sample with fruits

| | Image preprocessing method | | | | | |
| --- | --- | --- | --- | --- | --- | --- |
| | **Grayscale** | **Retinal Simula- tion Module** | **LoG filter** | **DoG filter** | **Roberts filter** | **MSRCR filter** |
| Average recognition accuracy at 10 launches | 78% | 87% | 80% | 80% | 71% | 76% |

## 6 DISCUSSION OF THE RESULTS OBTAINED

Table 2 shows that the same filters mirrored on the x-axis can lead to different results.

Several patterns have been observed:

1. Alternate colors on the main diagonal of the filter and the diagonals parallel to it.;

2. If there is no color alternation on the main diagonal of the filter and the diagonals parallel to it, the pattern recognition accuracy decreases.;

3. Alternating red and blue colors increases recognition accuracy more than alternating blue and green, red and green.

Using these patterns, it was possible, for example, to increase the accuracy for the filter shown in Figure 7.

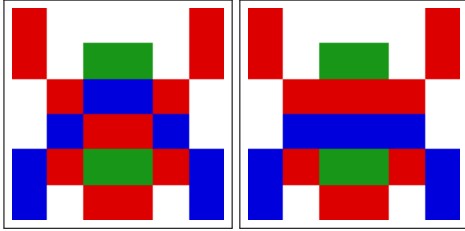

Figure 7: Fixed spider-like filter.

This filter had red and blue lines consisting of three pixels. After adding alternating red and blue colors in the center of the filter, the recognition accuracy increased from 79% to 85%.

Another example is the filters shown in Figure 8. Due to the repetition of colors on the diagonal, the filters showed the same pattern recognition accuracy of 77%.

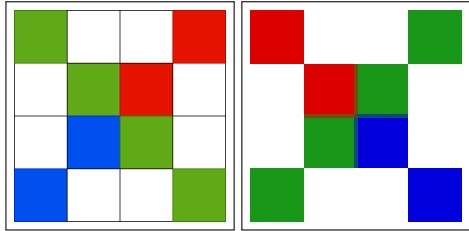

Figure 8: Filters similar to "X", showing the same accuracy.

Table 2: The results of the experiment

| Filters | Recognition accuracy at 10 launches |
|---|---|
|  | 73-80% |
|  | 81-83% |
|  | 84-86% |
|  | 87-89% |
|  | 90-92% |

This behavior of the module when pixels of different colors are located on the main diagonal and diagonals parallel to it is due to the fact that ON-cells are located at these positions. This hypothesis needs to be tested by a large number of experiments.

## 7 CONCLUSION

The work evaluated the effect of the location of receptors in the retinal simulation module on the ability of a neural network to recognize images. Filter options have been created based on existing camera filters and macaque retina images. Using the retinal simulation module, samples were prepared on which the experiment of this study was conducted.

Based on the results obtained, three patterns were observed. A hypothesis has been proposed that may explain these patterns. The hypothesis needs to be tested by a large number of experiments.

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
