# OpenReview forum: "Evaluation of the accuracy of pattern recognition by a neural network with various filters in the receptor layer of the retinal simulation module"
_ICLR.cc/2026/Conference — ICLR 2026 Conference Desk Rejected Submission_

### Official Review · Reviewer_rMAN · 2025-10-30

**Soundness:** 2
**Presentation:** 2
**Contribution:** 3
**Rating:** 4
**Confidence:** 4

**Summary:**

This paper explores the impact of receptor-layer filter arrangements in a bio-inspired retinal simulation module on neural network recognition accuracy. The authors construct a three-layer retinal simulator and test 94 filter configurations on a fruit image recognition task. The key finding is that diagonal red-blue alternating filter patterns boost fully-connected neural network (FCNN) accuracy to 92%, significantly outperforming traditional preprocessing methods. The authors attribute this phenomenon to the spatial distribution of retinal ON cells, proposing a biologically-inspired approach to image preprocessing.

**Strengths:**

1.	Testing 94 filter configurations far exceeds prior work in bio-inspired vision preprocessing, providing the first systematic analysis of spatial topology effects.
2.	ON/OFF cell modeling is well-founded in retinal neuroscience, and the link between diagonal patterns and ON-cell responses offers a plausible mechanism..
3.	The "red-blue diagonal alternation" principle  directly informs sensor design, with a +14% accuracy gain over baselines..

**Weaknesses:**

1.	Conclusions rely solely on FCNN – an obsolete architecture for vision tasks.  The critical absence of CNN/Transformer validation (e.g., ResNet, ViT) leaves open whether results generalize to modern models.
2.	Despite citing hexagonal lattices in biological retinas, all filters use rectangular grids.  This mismatch undermines the claimed bio-inspiration.
3.	The ON-cell distribution hypothesis lacks computational/biological support.  The conclusion section fails to restate this hypothesis, weakening theoretical contributions.
4.	Results on fruit images may overfit diagonal textures.  Tests on standard datasets (CIFAR/ImageNet) and comparisons with neuromorphic models (e.g., Spiking CNNs) are missing.

**Questions:**

1.	Why use FCNN instead of CNNs?  Can the conclusions hold for mainstream vision architectures?
2.	How does the rectangular grid assumption align with biological hexagonal lattices?  Will hexagonal filters alter the diagonal pattern effect?
3.	Have you tested generalization beyond fruit images (e.g., on CIFAR-10/100 or ImageNet subsets)?

---

### Official Review · Reviewer_vQiy · 2025-11-01

**Soundness:** 2
**Presentation:** 2
**Contribution:** 2
**Rating:** 2
**Confidence:** 4

**Summary:**

The paper proposes a novel set of preprocessing filters specifically tailored for image classification. These filters are inspired by the structure of the human retina and are used to simulate the distribution of photoreceptor cells. By applying these biologically motivated filters before feeding images into a neural network, the model can achieve improved feature extraction and higher recognition accuracy.

**Strengths:**

The paper provides new insights inspired by the visual processing mechanisms of the human eye, offering biologically motivated ideas for improving artificial vision systems.

**Weaknesses:**

The experiments presented in the paper are not sufficient to fully support the main proposal. In particular, the evaluation lacks comparative analysis with other existing models and architectures. To strengthen the validity of the findings, the authors should include experiments that compare their method against a wider range of baseline models and state-of-the-art approaches in image preprocessing and classification.

**Questions:**

Why did the authors use only a fruit image dataset for the experiments? This limited choice makes it difficult to evaluate the generality of the proposed method. Testing on additional and more diverse datasets would strengthen the validity of the results.

Could the authors clarify whether the retinal simulation module operates as a fixed preprocessing layer or if it is trainable (end-to-end with the network)?

How robust are the results to network architecture changes—e.g., CNNs or ResNets?

How does the module perform on standard benchmarks (e.g., CIFAR-10, ImageNet subsets)?

Are the “color alternation” effects observed consistent across multiple datasets and lighting conditions?

**Details Of Ethics Concerns:**

Why did the authors use only a fruit image dataset for the experiments? This limited choice makes it difficult to evaluate the generality of the proposed method.

---

### Official Review · Reviewer_oKn4 · 2025-11-01

**Soundness:** 2
**Presentation:** 2
**Contribution:** 1
**Rating:** 2
**Confidence:** 4

**Summary:**

The manuscript presents a three-layer retinal simulation module consisting of photoreceptors, bipolar cells, and ganglion cells, which is used solely as an image pre-processor. It investigates how different photoreceptor mosaics - drawing inspiration from Bayer, RGBW, X-Trans, and custom-designed patterns—impact classification accuracy on a 39-class fruit dataset with images sized at 100×100 pixels. A total of ninety-four module configurations are evaluated, keeping a fully connected classifier constant. The results, averaged over ten training sessions, indicate that the proposed module outperforms several classical pre-processing baselines. Additionally, qualitative observations reveal certain “rules” regarding colour alternation along the diagonals.

**Strengths:**

1. A clear and modular description of a biologically motivated pre-processor.
2. A systematic examination of various photoreceptor mosaics (total of 94 settings) with the classifier held constant, isolating the effect of pre-processing.

**Weaknesses:**

1. The idea is intriguing, and the empirical pattern regarding colour alternation shows promise. However, the current evaluation lacks robust baselines, generalisation tests, and thorough statistical analysis. Addressing these points would significantly strengthen the paper.
2. The abstract states the objective but would benefit from a concise research question. Moreover, the manuscript fails to discuss the prior work in relation to the current work in the discussion section. No limitation is mentioned either.

**Questions:**

1. The current classifier is a fully connected network that operates on raw pixel data. No convolutional baseline has been reported. This raises concerns about its practical utility, as modern vision systems typically utilise CNNs or ViTs, which are adept at learning colour and centre-surround structures. I recommend adding at least one small CNN (for example, a 3–5 layer convolutional network) and a lightweight ResNet-18 baseline, both with and without your proposed module. Please maintain the same training protocol as your current setup (SGD with Nesterov momentum, for 10 epochs, and repeated over 10 runs) to ensure fairness in comparison.
2. The details regarding the "fruit" dataset, including its source, licensing, and class list, are currently missing.
3. Please provide per-class accuracy, confusion matrices, calibration, and robustness against common corruptions. Additionally, show learning curves to determine whether the module accelerates convergence or merely shifts final accuracy.
4. This manuscript lacks an ablation experiment, which should include the following scenarios: (a) removing the ganglion stage; (b) using only the ON pathways or only the OFF pathways; (c) varying the receptive field radii; and (d) swapping the colour-channel assignments between L/M/S and RGB. Please report the contribution of each stage.

---

> ### Comment · Reviewer_oKn4 · 2025-11-25
>
> At this point, I have not received any responses from authors, so I do not recommend this paper for acceptance.

---

### Official Review · Reviewer_n3ZM · 2025-11-01

**Soundness:** 1
**Presentation:** 2
**Contribution:** 1
**Rating:** 2
**Confidence:** 4

**Summary:**

The paper investigates how different retinal receptor layouts, inspired by biological vision systems, affect image classification accuracy. The authors design a retinal simulation module that mimics the photoreceptor, bipolar, and ganglion layers of the human eye, using fixed filters to model center-surround receptive fields. Several receptor configurations are tested as preprocessing steps before a simple fully connected neural network trained on a 39-class fruit dataset. Results show that certain color arrangements, particularly alternating red and blue patterns, improve classification accuracy compared to standard preprocessing methods (grayscale, LoG, DoG). The article concludes with design guidelines for effective receptor layouts and suggests that biologically inspired mosaics can modestly enhance recognition performance.

**Strengths:**

The paper’s strength lies in its original, biologically inspired framing, which models retinal receptor mosaics as a preprocessing mechanism for image classification. It is methodically thorough, testing 94 receptor layouts under consistent training conditions, and presents precise, reproducible results. The idea offers a cross-disciplinary perspective linking computational vision and neurobiology, and the paper’s structure and visualizations make the findings reasonably clear.

**Weaknesses:**

The paper’s main weaknesses lie in its limited methodological rigor and narrow experimental scope. Relying on a single small dataset (fruit images, 100×100 pixels) and a simple fully connected network prevents meaningful generalization. There is no evaluation on standard benchmarks such as CIFAR or ImageNet, nor with modern architectures like ResNet, ConvNeXt, or Vision Transformers, so the reported improvements lack credibility and broader validation. The retinal simulation module also contains white or blank patches in the receptor layout, elements that do not exist in real Bayer filters or biological retinas—making the model optically and biologically implausible. These “white cells” distort spatial sampling density and frequency response, undermining fair comparison with true color filter arrays. Figure 6 should more clearly illustrate the proposed architecture. Its current presentation is sloppy and confusing, lacking structure and visual clarity. The authors are encouraged to redraw it using standard diagramming tools to produce a cleaner, more professional visualization of the model. The paper further suffers from weak theoretical grounding and the absence of statistical analysis; no uncertainty estimates, variance measures, or significance tests are provided. The discussion remains purely descriptive, offering unvalidated hypotheses without analytical support. To improve, the authors should strengthen the biological and optical modeling, expand experimental validation to diverse datasets and modern networks, and provide rigorous statistical and theoretical analysis.

**Questions:**

1.	Have you evaluated the proposed retinal simulation module on other datasets or architectures, such as ResNet, ConvNeXt, or Vision Transformers? Results on a single small fruit dataset limit the generalizability of your conclusions.

	2.	The receptor layout includes white or blank cells, which are not present in real Bayer filters or biological retinas. What is the rationale behind this design choice, and how does it affect the model’s optical and biological validity?

	3.	The reported accuracy differences are not supported by statistical analysis. Could you include standard deviations, confidence intervals, or significance tests across the ten runs to show that these differences are meaningful?

	4.	Figure 6 does not clearly illustrate the proposed architecture. Can you provide a higher-quality, properly labeled diagram showing the module structure, layer arrangement, and data flow?

	5.	Could you better relate your work to previous computational and biologically inspired vision models, such as color filter array optimization or neuromorphic vision front ends? This would clarify the novelty of your approach.

	6.	The observed performance trends (for example, alternating red and blue patterns) are only described heuristically. Can you provide an analytical or theoretical explanation for why these arrangements improve accuracy?

	7.	You mention that the algorithm for the bipolar layer has a time complexity of O(n⁴). Could you quantify its computational cost relative to standard convolutional preprocessing or CNN feature extraction?

	8.	Will you make your code and receptor configurations publicly available? Providing access to your implementation would help others replicate and verify your results.

---

### Note · Program_Chairs · 2026-01-17
**Submission Desk Rejected by Program Chairs**

The following references in this submission do not refer to real documents and/or have major errors in bibliographic information:

 - I. Izmailov, E. Sokolov, and A. Chernorizov. Psychophysiology of color vision. Publishing House of Moscow State University, 1989.
- P. Tupileikina. Research of the current state and possible prospects in the field of artificial intelligence. Scientific Research of the XXI Century, 26:41-44, 2023.